# IDEAL-RAG: Instruction-driven Dual-standpoint Elicitation and Alignment Linking for Retrieval Augmented Generation

## Abstract

Retrieval-augmented generation (RAG) equips large language models (LLMs) with external evidence, yet even minor retrieval noise or adversarial edits can override parametric knowledge and trigger hallucinations. Prior work mainly denoises contexts; far fewer methods explicitly balance internal memory with retrieved text. We present IDEAL-RAG, a three-stage, instruction-driven framework that (i) elicits latent knowledge, (ii) forms independent standpoints from internal memory and retrieved passages, and (iii) cross-checks them to produce a traceable rationale—without modifying retrievers or requiring additional labels. Across standard open-domain QA settings, IDEAL-RAG matches strong baselines on clean retrieval and, under adversarial counterfactual contexts, improves exact-match by up to +22.8% while roughly halving accuracy loss. Mechanistic analyses explain the gains: a Counterfactual Sensitivity Score (CSS) shows smaller confidence swings, and a layer-wise Parametric Knowledge Score (PKS) reveals steadier reliance on internal memory; ablations further identify parametric-knowledge elicitation as the primary driver of robustness. These results indicate that deliberate negotiation between what an LLM knows and what it reads yields more dependable RAG systems.

## 1 Introduction

Large language models (LLMs) excel at natural language generation (Brown et al., 2020; Team et al., 2023; Touvron et al., 2023; Bubeck et al., 2023), yet they often fail on recent events, rare entities, or domain-specific knowledge, producing hallucinations or refusals (Roberts et al., 2020; Dhingra et al., 2022; Jiang et al., 2023; Yu et al., 2023; Zhao et al., 2023; Wu et al., 2024). Since continuously retraining to cover all knowledge domains is infeasible, *Retrieval-Augmented Generation* (RAG) (Chen et al., 2017; Gao et al., 2023; Guu et al., 2020; Izacard et al., 2023b; Lewis et al., 2020) augments LLMs with retrieved documents, ideally grounding answers in verifiable evidence while still leveraging parametric memory.

However, this assumption proves fragile. Even minimal retrieval noise—irrelevant hits, partial matches, or adversarial edits (RAG noise)—can flip correct answers into confident hallucinations (Fang et al., 2024; Yoran et al., 2024; Yu et al., 2024; Li et al., 2023; Cuconasu et al., 2024). Studies show LLMs tend to over-rely on retrieved passages while underutilizing internal knowledge (Wadhwa et al., 2024; Sun et al., 2025). This has motivated a growing body of work on noise-robust RAG. Proposed defenses include requiring justification before answering (Yu et al., 2024), aggregating answers from subsets (Xiang et al., 2024), or combining reflection with self-consistency voting (Asai et al., 2023; Schulman et al., 2017; Ouyang et al., 2022). A lightweight variant, InstructRAG (Wei et al., 2025), achieves strong accuracy by prompting models to generate rationales, yet remains highly vulnerable to noise (Sun et al., 2025) and therefore serves as a standard baseline.

To evaluate robustness, researchers have introduced benchmarks that inject controlled noise, such as token-level hallucination corpora (Wu et al., 2024), irrelevant or misleading sentences in QA datasets (Yoran et al., 2024; Yang et al., 2018; Kwiatkowski et al., 2019), or paragraph-level replacements and counterfactual edits (Zhang et al., 2024; Fang et al., 2024). These stress tests consistently reveal brittleness in standard RAG pipelines.

Meanwhile, relatively little attention has been paid to the role of an LLM's own **parametric memory**. Conditional retrieval heuristics (Xu et al., 2023; Mallen et al., 2023; Jeong et al., 2024) or fusion-based methods (Wang et al., 2024a) attempt to balance sources but lack mechanisms for principled conflict resolution. Probing studies (Sun et al., 2025) reveal that mainstream RAG architectures increasingly suppress the use of parametric memory in order to curb hallucinations, thereby leaning almost entirely on retrieved evidence. While this strategy reduces uncontrolled reliance on internal knowledge, it simultaneously magnifies the system's vulnerability: any noise or adversarial corruption in the retrieval can dominate the generation process and severely compromise robustness.

This gap motivates our central question: **How can we leverage what a model already "knows" to remain robust when retrieval is incomplete or misleading?**

We address this by introducing IDEAL-RAG (Instruction-driven Dual-standpoint Elicitation and Alignment Linking), a three-stage framework that (i) explicitly elicits the model's internal knowledge, (ii) derives independent standpoints from both internal and external sources, and (iii) links them into a unified rationale. As illustrated in Figure 1, IDEAL-RAG balances intrinsic and retrieved knowledge, avoiding spurious anchoring on noisy passages. Experiments across multiple QA benchmarks and counterfactual settings demonstrate that IDEAL-RAG sustains competitive accuracy under clean retrieval while substantially improving robustness against corrupted evidence.

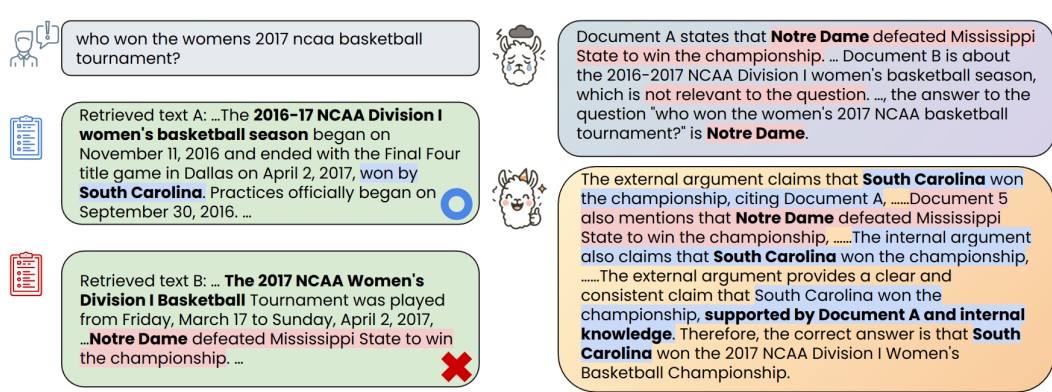

Figure 1: The figure contrasts a standard retrieval-augmented baseline with IDEAL-RAG under a noisy-retrieval scenario. Whereas the baseline model (upper-right) anchors on misleading context and produces an incorrect rationale, IDEAL-RAG (lower-right) draws on its internal knowledge, balances conflicting sources, and delivers a stable, correct answer, demonstrating its robustness to retrieval noise.

## 2  METHODOLOGY: IDEAL-RAG

Large language models (LLMs) excel at following instructions, preserving style, and composing multi-step explanations with minimal supervision. Prior work shows that with carefully curated exemplars, models can acquire sophisticated behaviors without heavy annotation or rewards (Brown et al., 2020; Asai et al., 2023; Wei et al., 2025). Building on this, we present IDEAL-RAG (Instruction-Driven Evidence Alignment and Linking), a three-stage framework that contrasts an LLM's parametric knowledge with retrieved passages and then reconciles them.

### 2.1  MOTIVATION

Existing RAG systems often treat internal knowledge as secondary. However, deciding when to trust memory versus retrieved text is non-trivial, especially under noisy or adversarial retrieval. Some methods suppress parametric knowledge and lean almost entirely on external sources, but real deployments cannot assume perfect retrieval. Our design instead (i) explicitly elicits what the model already "knows," (ii) requires independent standpoints from both sources, and (iii) introduces a linking stage to reconcile conflicts. This separation prevents premature fusion and encourages transparent reasoning. A high-level overview is shown in Figure 2.

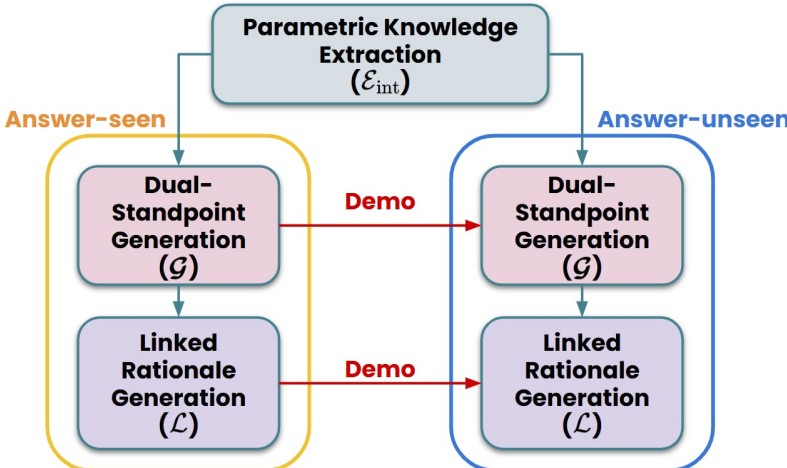

Figure 2: High-level view of IDEAL-RAG. The pipeline first elicits parametric knowledge ($\mathcal{E}_{\text{int}}$), then runs two stages in mirrored regimes. On a small answer-seen split, it generates internal/external standpoints ($\mathcal{G}$) and a linked rationale ($\mathcal{L}$) to build a demo bank. On answer-unseen questions, the same two stages are executed conditioned on these demos, yielding the final linked rationale and answer. This layout makes internal and retrieved evidence explicit and comparable before integration.

## 2.2 PROBLEM FORMULATION

We consider an open-domain corpus:

$$\mathcal{C} = \big\{(q_i, a_i, \mathcal{D}_i)\big\}_{i=1}^{N}, \tag{1}$$

where $q_i$ is a query, $a_i$ the gold answer, and $\mathcal{D}_i = \mathcal{R}(q_i) \subset \mathcal{T}$ passages retrieved by a frozen retriever $\mathcal{R}$ from text collection $\mathcal{T}$. The retriever is deliberately fixed to isolate generation-side robustness. Following prior work (Wei et al., 2025; Asai et al., 2023), we use exact-match accuracy as the primary evaluation metric. For a test set $\mathcal{C}_{\text{test}}$, EM is defined as:

$$\text{Acc} = \frac{1}{|\mathcal{C}_{\text{test}}|} \sum_{(q,a) \in \mathcal{C}_{\text{test}}} \mathbf{1}\Big[a \subseteq R, \ R = \text{IDEAL-RAG}(q, \mathcal{D})\Big], \tag{2}$$

where a prediction $R$ is counted as correct if any string in the reference answer set $a$ appears in the final output.

## 2.3 THREE-STAGE PIPELINE

### 2.3.1 PARAMETRIC KNOWLEDGE EXTRACTION ($\mathcal{E}_{\text{INT}}$)

Given a question $q$, we elicit the model's latent knowledge $K_{\text{int}}$ through structured prompting. This step surfaces internal evidence before consulting retrieved passages.

### 2.3.2 DUAL-SOURCE STANDPOINT GENERATION ($\mathcal{G}$)

We enforce two independent standpoints: one grounded in $K_{\text{int}}$ and the other in retrieved passages $\mathcal{D}$.

1. **Answer-Seen (Seed Construction).** On a small seed set $\mathcal{C}_{\text{seed}}$, we reveal the gold answer $a$ so the model can produce "ideal" reasoning trajectories. Internal and external standpoints $\mathcal{S}_{\text{int}}^{\star}, \mathcal{S}_{\text{ext}}^{\star}$ are stored in exemplar banks $\mathcal{B}_{\text{int}}, \mathcal{B}_{\text{ext}}$.

2. **Answer-Unseen (Inference).** For the remaining data, answers are hidden. Conditioned on exemplar banks, the model generates $\hat{\mathcal{S}}_{\text{int}}, \hat{\mathcal{S}}_{\text{ext}}$ via in-context learning. Each standpoint contains evidence, reasoning, and uncertainty notes.

### 2.3.3 LINKED RATIONALE GENERATION ($\mathcal{L}$)

The final step reconciles $(\hat{\mathcal{S}}_{\text{int}}, \hat{\mathcal{S}}_{\text{ext}})$ into a conflict-aware rationale.

1. **Answer-Seen Linking.** Seed examples are used to construct a third exemplar bank $\mathcal{B}_{\text{link}}$, capturing cross-examination behaviors.

2. **Answer-Unseen Linking.** At test time, $\mathcal{L}$ produces rationales by referencing $\mathcal{B}_{\text{link}}$ through few-shot inference.

3. **Optional Instruction Tuning.** In the SFT variant, linked rationales serve as training pairs to fine-tune the backbone model $\Theta_0$, yielding $\Theta_{\text{link}}$. This variant offers further gains, though ICL alone performs strongly. The explicit training objective is deferred to Appendix A.

### 2.3.4 IMPLEMENTATION NOTES

All modules operate on a frozen backbone $\Theta_0$ without external verifiers or retriever modifications. Ground-truth answers are used only in seed construction to populate exemplar banks. A comprehensive overview of the framework is presented in Figure 3. The full algorithmic details are provided in Appendix A, and the complete prompt templates are included in Appendix C.

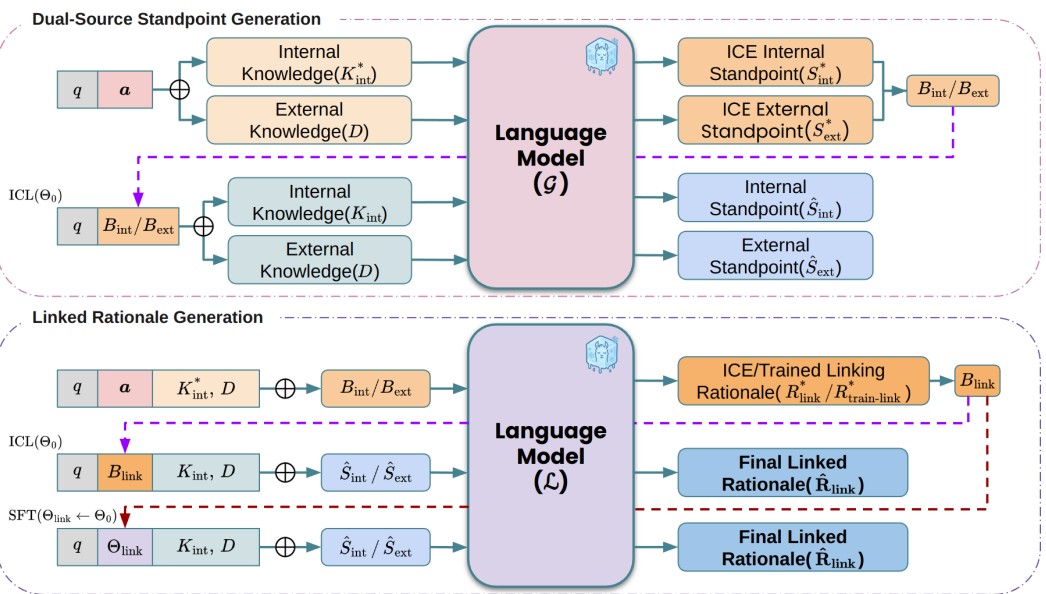

Figure 3: Overview of IDEAL-RAG. The system first constructs two standpoints: one derived from the model's internal memory, and the other from the retrieved passages. These standpoints serve as an interface that bridges parametric and non-parametric knowledge. The model then performs deliberative reasoning over the two to generate a unified rationale and final answer. This reasoning is implemented either via in-context prompting or through a lightweight fine-tuned prediction head. The full process comprises two stages: one where the answer is revealed (answer-seen) and one where it remains hidden (answer-unseen).

## 3 EXPERIMENTS

### 3.1 EXPERIMENTS SETTING

**Open-Domain QA Benchmarks and Metrics.** We test on four widely-used datasets with diverse reasoning requirements: PopQA(Mallen et al., 2022), Natural Questions (NQ)(Kwiatkowski et al.,

2019), TriviaQA(Joshi et al., 2017), and 2WikiMultiHopQA(Ho et al., 2020). Following prior work (Asai et al., 2023; Wang et al., 2024b; Wei et al., 2025), each query is paired with the top-$k$ passages from a hybrid retriever (BM25(Robertson & Walker, 1994), DPR(Karpukhin et al., 2020), Contriever(Izacard et al., 2023a)). This setup ensures comparability while reflecting realistic imperfect retrieval, where gold passages may be absent. Table 1 reports Recall@$k$, confirming that substantial portions of gold evidence remain unretrieved.

Performance is measured by Exact-Match (EM) accuracy (Eq. 2). To assess robustness beyond clean retrieval, we also consider three complementary metrics. The **Accuracy Degradation Ratio (ADR)** quantifies how much EM drops when clean passages are replaced with noisy or counterfactual ones, where a lower ADR indicates greater robustness. The **Counterfactual Sensitivity Score (CSS)** reflects how strongly the model's answer confidence fluctuates under adversarial edits, with smaller values corresponding to steadier reasoning. Finally, the **Parametric Knowledge Score (PKS)** measures the extent to which the model draws on its parametric memory (stored in weights) relative to retrieved evidence; stable PKS across clean and noisy conditions suggests a balanced reliance on internal and external knowledge. Formal definitions of ADR, CSS, and PKS are deferred to Appendix A.

**Counterfactual Test Sets.** Since real-world retrieval is rarely clean, we construct two stress-test suites by replacing gold answer spans with semantically similar but incorrect entities (Fang et al., 2024) (e.g., *"Barack Obama"→"Michelle Obama"*). In the **Counter-All** setting, every answer-containing passage is overwritten, while in the **Counter-Mix** setting, only half of the supporting passages are corrupted when multiple gold-containing passages exist. The number of applicable queries for each dataset is reported in Table 1, providing a systematic way to evaluate robustness under adversarial retrieval.

Table 1: Dataset statistics and retrieval setting (with counter values).

| Dataset | Train | Test | Retriever | Top-K | R@K | $\widetilde{\mathcal{D}}_{c\_m}$ | $\widetilde{\mathcal{D}}_{c\_a}$ |
|---|---|---|---|---|---|---|---|
| PopQA | 12,868 | 1,399 | Contriever | 5 | 68.7 | 578 | 961 |
| Natural Questions | 79,168 | 3,610 | DPR | 5 | 68.8 | 1,634 | 2,482 |
| TriviaQA | 78,785 | 11,313 | Contriever | 5 | 73.5 | 6,548 | 8,313 |
| 2WikiMultiHopQA | 167,454 | 12,576 | BM25 | 10 | 40.7 | 3,645 | 5,122 |

**Baselines.** To contextualize IDEAL-RAG's performance, we compare it against several representative systems spanning both training-free and trainable paradigms. As a training-free reference, RALM (Ram et al., 2023) simply concatenates the top-$k$ retrieved passages with the query and relies on the frozen language model to generate an answer. On the trainable side, we include a Vanilla SFT baseline, where the model is fine-tuned directly on retrieved contexts to maximize answer likelihood without any additional reasoning objectives. We also evaluate Self-RAG (Asai et al., 2023), which is a stronger baseline that integrates retrieval with dynamic reflection: the model decides when to retrieve, critiques both the passages and its own outputs using special "reflection tokens," and leverages these signals to improve factuality and citation accuracy. Finally, we include InstructRAG (Wei et al., 2025), a lightweight but highly competitive method that teaches models to generate rationales from retrieved evidence and is widely recognized for its robustness to noise. For fairness, results marked with $\star$ reflect the stronger of either the authors' original release or our faithful re-implementation. Note that InstructRAG was originally trained with full-parameter fine-tuning, whereas all our experiments—including IDEAL-RAG—employ parameter-efficient tuning.

## 3.2 MAIN RESULTS

Table 2 summarizes EM accuracy across clean and counterfactual settings. On clean corpora, IDEAL-RAG remains competitive with InstructRAG, trailing by modest margins (e.g., $-7.2\%$ on 2WikiMultiHopQA, $-1.69\%$ on Natural Questions). These differences align with parametric coverage: when answers are less frequently encoded in the base model (PopQA, 2WikiMultiHopQA), IDEAL-RAG underperforms; when coverage is richer, the gap narrows. Importantly, IDEAL-RAG often operates close to the empirical ceiling set by retrieval recall (Table 1), and in some cases even exceeds R@$k$ by leveraging internal memory—something retrieval-only baselines cannot achieve.

Table 2: Exact-match accuracy (%) on four QA benchmarks. Columns report clean retrieval results (Origin) as well as performance under two counterfactual corruption settings—50% passage edits ($\widetilde{\mathcal{D}}_{c\_m}$) and full-passage edits ($\widetilde{\mathcal{D}}_{c\_a}$). The upper block shows prompt-only models; the lower block includes models further fine-tuned with LoRA. vanilla[†] scores are taken from Wei et al. (2025). Best numbers per column are **bold**.

| Method | PopQA | | | NQ | | | TriviaQA | | | MultiHopQA | | |
|---|---|---|---|---|---|---|---|---|---|---|---|---|
| | Origin | $\widetilde{\mathcal{D}}_{c\_m}$ | $\widetilde{\mathcal{D}}_{c\_a}$ | Origin | $\widetilde{\mathcal{D}}_{c\_m}$ | $\widetilde{\mathcal{D}}_{c\_a}$ | Origin | $\widetilde{\mathcal{D}}_{c\_m}$ | $\widetilde{\mathcal{D}}_{c\_a}$ | Origin | $\widetilde{\mathcal{D}}_{c\_m}$ | $\widetilde{\mathcal{D}}_{c\_a}$ |
| | *w/o Training* | | | | | | | | | | | |
| RALM | 61.97 | 72.66 | 32.36 | 56.37 | 69.34 | 25.38 | 71.47 | 82.48 | 42.80 | 43.37 | 60.81 | 54.98 |
| InstructRAG | **63.97** | 78.55 | 40.69 | **62.52** | 77.36 | 25.10 | **76.95** | 89.80 | 53.69 | **49.27** | **79.75** | 59.59 |
| **IDEAL-RAG** | 62.76 | **81.14** | **46.51** | 60.83 | **77.97** | **51.97** | 76.82 | **91.95** | **78.73** | 47.60 | 76.76 | **60.82** |
| | *w/ Training* | | | | | | | | | | | |
| vanilla[†] | 61.00 | – | – | 56.60 | – | – | 73.90 | – | – | 43.8 | – | – |
| Self-RAG* | 52.47 | 65.57 | 25.70 | 40.17 | 48.04 | 10.39 | 64.39 | 73.09 | 45.06 | 23.40 | 37.96 | 27.24 |
| InstructRAG* | **65.90** | **89.45** | **49.32** | **65.68** | 80.29 | 32.67 | **78.70** | 90.79 | 57.80 | **57.19** | **87.02** | **66.26** |
| **IDEAL-RAG** | 64.05 | 84.08 | 47.76 | 63.71 | **80.97** | **55.48** | 77.19 | **92.21** | **78.58** | 50.01 | 80.85 | 64.31 |

When noise is introduced, IDEAL-RAG demonstrates clear robustness. Under $\widetilde{\mathcal{D}}_{c\_m}$, it matches or slightly surpasses the strongest baselines. With full corruption ($\widetilde{\mathcal{D}}_{c\_a}$), IDEAL-RAG achieves large gains—up to +22.8% on Natural Questions and +26.9% in training-free settings—with consistent improvements on PopQA, TriviaQA, and 2WikiMultiHopQA. A caveat emerges in benchmarks with limited internal coverage: trained InstructRAG can occasionally surpass IDEAL-RAG under full corruption, as reliance on memory becomes a liability when little relevant information is stored. Detailed answer-containment analysis in § B.2 supports this observation.

Overall, these results confirm that explicitly surfacing and reconciling internal knowledge preserves competitive clean accuracy while yielding substantial resilience to retrieval noise.

## 3.3 ACCURACY DEGRADATION RATIO (ADR)

Table 3 shows that IDEAL-RAG consistently achieves the lowest ADR across all datasets and corruption settings. On Natural Questions with $\widetilde{\mathcal{D}}_{c\_a}$, InstructRAG suffers a 70.61% accuracy drop, while IDEAL-RAG limits the decline to 35.22%—nearly halving degradation. Similar improvements are observed on PopQA, TriviaQA, and 2WikiMultiHopQA. These results demonstrate that IDEAL-RAG not only narrows accuracy gaps in noisy settings but also retains a larger portion of its clean-data competence, underscoring its practical reliability.

## 3.4 INTERNAL-MECHANISM ANALYSIS

### 3.4.1 COUNTERFACTUAL SENSITIVITY SCORE (CSS)

When adversarial passages are introduced, IDEAL-RAG maintains notably steadier answer probabilities compared to InstructRAG. On Natural Questions and TriviaQA, its predictions remain tightly concentrated, whereas InstructRAG displays wide, heavy-tailed distributions with frequent confidence swings. For instance, under the fully adversarial $\widetilde{\mathcal{D}}_{c\_a}$ setting, IDEAL-RAG's average CSS is just 0.830, while InstructRAG spikes to 3.657. These results (visualized in Figure 4) reveal that IDEAL-RAG's explicit knowledge surfacing prevents abrupt shifts in belief, aligning with the ADR findings from § 3.3.

### 3.4.2 PARAMETRIC KNOWLEDGE SCORE (PKS)

A second perspective comes from examining how much each transformer block injects parametric knowledge when retrieval is corrupted. Both models show increased PKS under noise, but IDEAL-

Table 3: ADR ($\downarrow$) measures accuracy drop from clean to corrupted retrieval. We compare methods on four QA benchmarks under partial ($\widetilde{\mathcal{D}}_{c\_m}$) and full ($\widetilde{\mathcal{D}}_{c\_a}$) passage corruption. IDEAL-RAG consistently shows the lowest degradation, both with and without fine-tuning.

| Method | PopQA | | NQ | | TriviaQA | | MultiHopQA | |
|---|---|---|---|---|---|---|---|---|
| | $\widetilde{\mathcal{D}}_{c\_m}$ | $\widetilde{\mathcal{D}}_{c\_a}$ | $\widetilde{\mathcal{D}}_{c\_m}$ | $\widetilde{\mathcal{D}}_{c\_a}$ | $\widetilde{\mathcal{D}}_{c\_m}$ | $\widetilde{\mathcal{D}}_{c\_a}$ | $\widetilde{\mathcal{D}}_{c\_m}$ | $\widetilde{\mathcal{D}}_{c\_a}$ |
| | | | | *w/o Training* | | | | |
| RALM | 20.46 | 63.33 | 18.83 | 67.83 | 13.11 | 53.16 | 4.49 | 21.47 |
| InstructRAG | 16.84 | 54.95 | 15.05 | 70.61 | 8.26 | 43.60 | 6.26 | 22.71 |
| **IDEAL-RAG** | **13.95** | **47.35** | **10.60** | **35.22** | **4.49** | **15.74** | **4.44** | **15.15** |
| | | | | *w/ Training* | | | | |
| Self-RAG* | 21.86 | 65.89 | 27.59 | 81.72 | 15.58 | 45.08 | 12.25 | 27.90 |
| InstructRAG* | 11.53 | 50.68 | 14.64 | 63.15 | 7.34 | 39.77 | **4.03** | 21.84 |
| **IDEAL-RAG** | **11.64** | **46.76** | **9.56** | **33.64** | 4.64 | 16.27 | 4.50 | **15.67** |

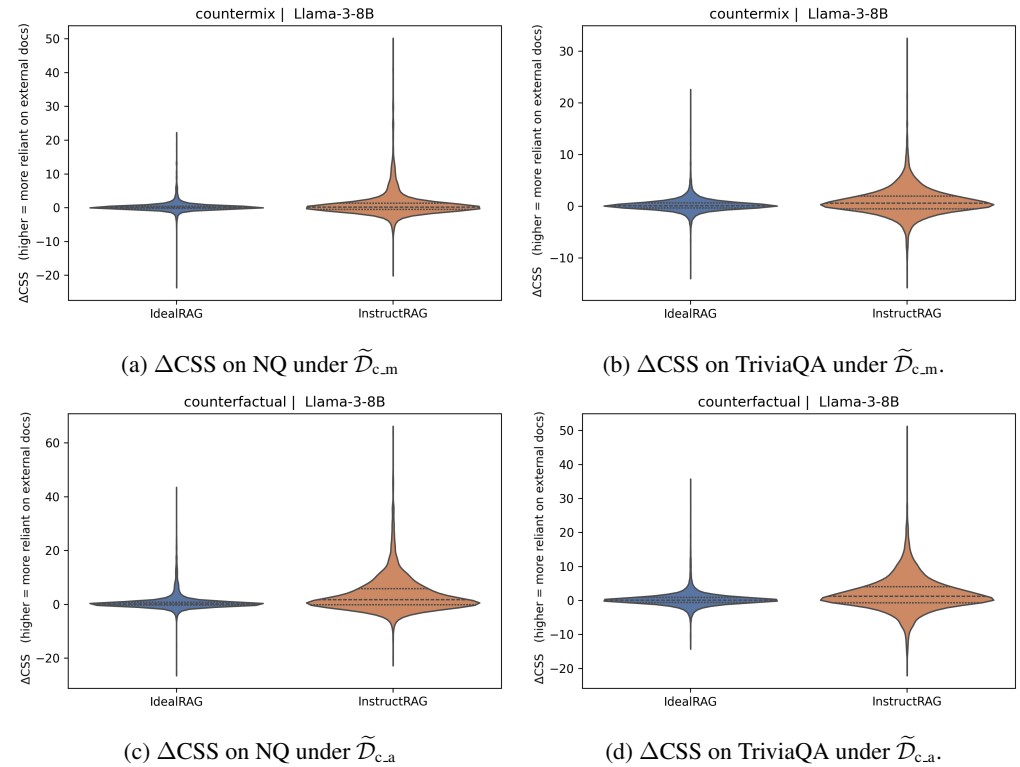

(a) $\Delta$CSS on NQ under $\widetilde{\mathcal{D}}_{c\_m}$

(b) $\Delta$CSS on TriviaQA under $\widetilde{\mathcal{D}}_{c\_m}$.

(c) $\Delta$CSS on NQ under $\widetilde{\mathcal{D}}_{c\_a}$

(d) $\Delta$CSS on TriviaQA under $\widetilde{\mathcal{D}}_{c\_a}$.

Figure 4: Violin plots compare changes in answer-token log-probabilities between the clean set and (a, b) the mixed-counterfactual set ($\widetilde{\mathcal{D}}_{c\_m}$) or (c, d) the fully counterfactual set ($\widetilde{\mathcal{D}}_{c\_a}$) on Natural Questions and TriviaQA. IDEAL-RAG (blue) shows narrower, lower-centered violins, indicating stable confidence, while InstructRAG (orange) displays wider, higher-centered violins, reflecting stronger reliance on corrupted evidence.

RAG's shifts are consistently smaller, reflecting its early extraction of internal knowledge rather than reactive reliance on the FFN pathway (Figure 5). This leads to a more stable residual stream and reduces hallucination risk.

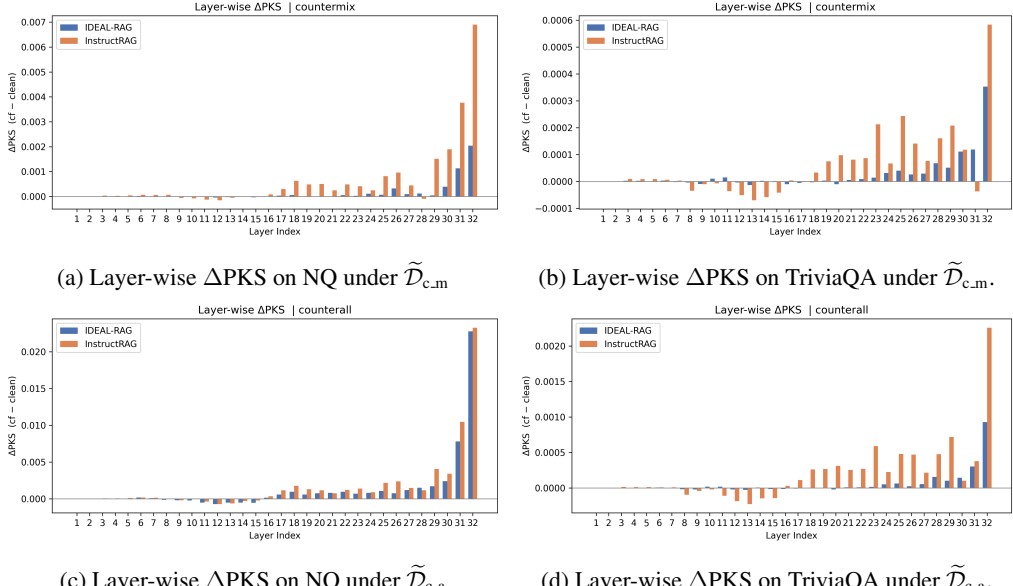

(a) Layer-wise $\Delta$PKS on NQ under $\widetilde{\mathcal{D}}_{\text{c\_m}}$

(b) Layer-wise $\Delta$PKS on TriviaQA under $\widetilde{\mathcal{D}}_{\text{c\_m}}$.

(c) Layer-wise $\Delta$PKS on NQ under $\widetilde{\mathcal{D}}_{\text{c\_a}}$

(d) Layer-wise $\Delta$PKS on TriviaQA under $\widetilde{\mathcal{D}}_{\text{c\_a}}$.

Figure 5: For each transformer block we plot the rise in PKS after replacing the clean passages with counterfactual ones. Orange bars are IDEAL-RAG, blue bars InstructRAG. (a, b) show the 50 % corruption mix ($\widetilde{\mathcal{D}}_{\text{c\_m}}$) on Natural Questions and TriviaQA; (c, d) show the full-corruption setting ($\widetilde{\mathcal{D}}_{\text{c\_a}}$). IDEAL-RAG requires only modest additional parametric input at deeper layers and maintains stable behavior, whereas InstructRAG exhibits larger spikes, indicating a late, reactive fallback to internal memory when retrieval is unreliable.

More detailed outcome-conditioned analyses, which compare correct versus incorrect predictions and contrast IDEAL-RAG with InstructRAG, are deferred to Appendix B.1. In summary, IDEAL-RAG's mechanism of proactively structuring parametric evidence leads to both more stable CSS and more interpretable PKS dynamics. This mechanistic consistency explains the aggregate accuracy gains reported in Table 2, confirming that structured dual-standpoint reasoning not only improves performance but also produces more predictable internal behavior.

## 4 ANALYSIS

### 4.1 ABLATION STUDY

To isolate the contributions of each module in IDEAL-RAG, we evaluate two reduced variants. The first removes both parametric extraction and standpoint generation, essentially collapsing the pipeline into InstructRAG's one-shot recipe (**w/o $\mathcal{E}_{\text{int}}$ and $\mathcal{G}$**). The second retains explicit extraction of internal knowledge but skips separate standpoint generation, feeding the question, passages, and elicited memory directly into the fusion step (**w/o $\mathcal{G}$**). The detailed prompt settings for these ablations are provided in Appendix C.

Results on Natural Questions and TriviaQA (Table 4) highlight two findings. First, **parametric extraction is decisive**: without explicit extraction and standpoints, robustness collapses under full counterfactual noise, with EM drops of -21.2% on Natural Questions and -25.7% on TriviaQA, even though performance on clean and partially noisy sets remains relatively stable. This confirms that proactive elicitation of parametric memory is the core driver of resilience. Second, **standpoint generation matters but is complementary**: removing this stage results in only modest EM declines (2–3%) across settings. While not as critical as extraction, standpoints help resolve residual conflicts and improve justification quality, acting as a stabilizer.

Table 4: Columns report performance on the original retrieval context and on the two counterfactual test suites—$\widetilde{\mathcal{D}}_{c\_m}$ and $\widetilde{\mathcal{D}}_{c\_a}$. Rows progressively omit key IDEAL-RAG stages: (i) both parametric-knowledge extraction and standpoints, (ii) standpoints only, and (iii) the full model.

| | NQ | | | TriviaQA | | |
|---|---|---|---|---|---|---|
| Method | Origin | $\widetilde{\mathcal{D}}_{c\_m}$ | $\widetilde{\mathcal{D}}_{c\_a}$ | Origin | $\widetilde{\mathcal{D}}_{c\_m}$ | $\widetilde{\mathcal{D}}_{c\_a}$ |
| w/o $\mathcal{E}_{int}$&$\mathcal{G}$ | 61.77 | 79.74(↑1.77%) | 30.74(↓21.23%) | 76.05 | 90.13(↓2.08%) | 52.85(↓25.73%) |
| w/o $\mathcal{G}$ | 60.86 | 76.99(↓0.98%) | 48.71(↓3.26%) | 76.07 | 92.20(↓0.01%) | 76.68(↓1.90%) |
| w/ all | 60.83 | 77.97 | 51.97 | 77.19 | 92.21 | 78.58 |

## 4.2 Training Data Analysis

We fine-tune both IDEAL-RAG and InstructRAG on 5,000 counterfactual training instances from Natural Questions (Table 5). Adding noise improves InstructRAG's robustness but reduces its clean accuracy. In contrast, IDEAL-RAG improves on both clean and noisy sets, still outperforming InstructRAG under corruption. This suggests IDEAL-RAG's gains stem from its architecture rather than data-specific effects.

Table 5: Natural-Questions results under three training regimes. Each block reports accuracy on the clean set (Original, which meets the counterfactual construction criteria) as well as on two counterfactual test sets ($\widetilde{\mathcal{D}}_{c\_m}$, $\widetilde{\mathcal{D}}_{c\_a}$). The corresponding Answer Degradation Rate (ADR) is also included.

| Method | $Origin_{c\_m}$ | $\widetilde{\mathcal{D}}_{c\_m}$ | ADR($\widetilde{\mathcal{D}}_{c\_m}$) | $Origin_{c\_a}$ | $\widetilde{\mathcal{D}}_{c\_a}$ | ADR($\widetilde{\mathcal{D}}_{c\_a}$) |
|---|---|---|---|---|---|---|
| | *Training w/ normal data* | | | | | |
| InstructRAG* | **94.06** | 80.29 | 14.64 | **88.68** | 32.68 | 63.15 |
| **IDEAL-RAG** | 89.53 | **80.97** | **9.56** | 83.6 | **55.48** | **33.64** |
| | *Training w/ counter_mix data* | | | | | |
| InstructRAG | **90.7** | 81.33 | 10.33 | 83.16 | 30.7 | 63.08 |
| **IDEAL-RAG** | 89.84 | **81.88** | **8.86** | **83.56** | **54.47** | **34.81** |
| | *Training w/ counter_all data* | | | | | |
| InstructRAG | **90.7** | **83.11** | 8.37 | 83.48 | 37.27 | 55.35 |
| **IDEAL-RAG** | 90.02 | 82.86 | **7.95** | **83.96** | **55.56** | **33.83** |

## 5 Conclusion

This work revisits retrieval-augmented generation from the perspective of the LLM's own parametric memory. We introduced IDEAL-RAG, a three-stage framework that elicits internal knowledge, develops independent standpoints from internal and external sources, and links them into a unified rationale. Experiments across four QA benchmarks show that IDEAL-RAG maintains competitive clean performance while substantially reducing degradation under counterfactual noise. Mechanistic analyses further confirm its stability, with CSS revealing reduced confidence swings and PKS indicating steadier reliance on parametric memory, while ablations highlight parametric extraction as the decisive driver of robustness. These findings demonstrate that deliberate negotiation between what an LLM knows and what it reads offers a principled path toward more dependable RAG systems and opens avenues for extending this negotiation framework to longer contexts, adaptive retrieval, and multi-step reasoning beyond QA.

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

# A IMPLEMENTATION DETAILS

## A.1 FULL ALGORITHM

The complete pseudocode for IDEAL-RAG is provided in Algorithm 1.

---

**Algorithm 1 IDEAL-RAG**

---

**Require:** QA corpus $\mathcal{C} = \{(q_i, a_i, \mathcal{D}_i)\}$ (including train and test); frozen backbone $\Theta_0$
**Ensure:** Fusion weights $\Theta_{\text{fuse}}$; at test time $\hat{\mathcal{R}}_{\text{fuse}}$
    **Standpoint Generation ($k \ll |\mathcal{C}|$):**
1: **for all** $(q, a, \mathcal{D}) \in \mathcal{C}_{\text{seed}}$ **do**
2:     $K_{\text{int}}^\star \leftarrow \mathcal{E}_{\text{int}}(q; \Theta_0)$                                 ▷ Extracted Parametric Knowledge
3:     $\mathcal{S}_{\text{int}}^\star \leftarrow \mathcal{G}(q, a, K_{\text{int}}^\star; \Theta_0)$                              ▷ answer-*seen*
4:     $\mathcal{S}_{\text{ext}}^\star \leftarrow \mathcal{G}(q, a, \mathcal{D}; \Theta_0)$
5:     add $\mathcal{S}_{\text{int}}^\star$ to internal exemplar bank $\mathcal{B}_{\text{int}}$
6:     add $\mathcal{S}_{\text{ext}}^\star$ to external exemplar bank $\mathcal{B}_{\text{ext}}$
7: **for all** $(q, a, \mathcal{D}) \in \mathcal{C} \setminus \mathcal{C}_{\text{seed}}$ **do**          ▷ Standpoint Generator (both test and train)
8:     $K_{\text{int}} \leftarrow \mathcal{E}_{\text{int}}(q; \Theta_0)$                                   ▷ answer-*unseen*
9:     $\hat{\mathcal{S}}_{\text{int}} \leftarrow \mathcal{G}(q, K_{\text{int}}; \Theta_0, \text{ICE} = \mathcal{B}_{\text{int}})$
10:     $\hat{\mathcal{S}}_{\text{ext}} \leftarrow \mathcal{G}(q, \mathcal{D}; \Theta_0, \text{ICE} = \mathcal{B}_{\text{ext}})$

    **Linked Rationale Generation:**
1: **for all** $(q, a, \mathcal{D}) \in \mathcal{C}_{\text{seed}}$ **do**
2:     $\mathcal{R}_{\text{link}}^\star \leftarrow \mathcal{L}(q, a, \mathcal{D}, K_{\text{int}}^\star, \mathcal{S}_{\text{int}}^\star, \mathcal{S}_{\text{ext}}^\star; \Theta_0)$           ▷ answer-*seen*
3:     add $\mathcal{R}_{\text{link}}^\star$ to integrated exemplar bank $\mathcal{B}_{\text{link}}$
4: **for all** $(q, a, \mathcal{D}) \in \mathcal{C} \setminus \mathcal{C}_{\text{seed}}$ **do**          ▷ Linking-based Integrator
5:     **if** MODE==IN-CONTEXT LEARNING **then**
6:         $\hat{\mathcal{R}}_{\text{link}} \leftarrow \mathcal{L}((q, \mathcal{D}, K_{\text{int}}, \hat{\mathcal{S}}_{\text{int}}, \hat{\mathcal{S}}_{\text{ext}}) \in \mathcal{C}_{\text{test}}; \Theta_0, \text{ICE} = \mathcal{B}_{\text{link}})$    ▷ answer-*unseen*
7:     **else if** MODE==FINE-TUNING **then**
8:         $\mathcal{R}_{\text{train-link}}^\star \leftarrow \mathcal{L}((q, a, \mathcal{D}, K_{\text{int}}, \hat{\mathcal{S}}_{\text{int}}, \hat{\mathcal{S}}_{\text{ext}}) \in \mathcal{C}_{\text{train}}; \Theta_0)$     ▷ answer-*seen*

9:         $\Theta_{\text{link}} \leftarrow Update(\mathcal{R}_{\text{link-train}}^\star | ((q, \mathcal{D}, \hat{\mathcal{S}}_{\text{int}}, \hat{\mathcal{S}}_{\text{ext}}) \in \mathcal{C}_{\text{train}}; \Theta_0))$
10:         $\hat{\mathcal{R}}_{\text{link}} \leftarrow \mathcal{L}((q, \mathcal{D}, K_{\text{int}}, \hat{\mathcal{S}}_{\text{int}}, \hat{\mathcal{S}}_{\text{ext}}) \in \mathcal{C}_{\text{test}}; \Theta_{\text{link}})$       ▷ answer-*unseen*
11: **return** $\hat{\mathcal{R}}_{\text{link}}$

---

## A.2 TRAINING, INFERENCE, AND RETRIEVER DETAILS

All models are built on LLAMA-3-8B-Instruct. Fine-tuning uses LoRA(Hu et al., 2021) (rank 8, $\alpha$ 16, dropout 0.05) with two epochs, cosine-decayed AdamW(Loshchilov & Hutter, 2017) at $2.5 \times 10^{-5}$, warm-up 3 %, and a global batch of one million tokens accumulated on two A100-80 GB GPUs (DeepSpeed ZeRO-2[(Rajbhandari et al., 2020)], bf16). We retain just ten thousand randomly chosen training questions per set because the loss plateaus quickly. Inference is done with vLLM(Kwon et al., 2023) in greedy mode; following the InstructRAG recipe, we include two exemplars per prompt when ICL is required.

For retrieval, we adopted the Wikipedia snapshot released by Karpukhin et al. (2020), segmented into fixed-length passages ($\leq$100 tokens). Different datasets were paired with retrievers optimized for their domain: Contriever-MS MARCO (Izacard et al., 2023a) for PopQA and TriviaQA, DPR (Karpukhin et al., 2020) for NQ, GTR (Ni et al., 2021) for ASQA, and BM25 (Robertson & Walker, 1994) via Pyserini (Lin et al., 2021) for 2WikiMultiHopQA. The retrieval depth was set to 5 passages per query, except for multi-hop tasks where 10 passages were used. Official checkpoints were employed for all dense retrievers, ensuring consistency with prior work (Asai et al., 2023; Ram et al., 2023).

## A.3 Evaluation Metrics

**Accuracy Degradation Ratio (ADR).** While EM measures overall performance, it does not capture robustness under noisy retrieval. We therefore introduce ADR, which quantifies the fraction of accuracy lost when clean passages are replaced by counterfactual variants $cf \in \widetilde{\mathcal{D}}_{\text{c\_a}}, \widetilde{\mathcal{D}}_{\text{c\_m}}$:

$$\text{ADR}_x = \frac{\text{EM}_{\text{clean}} - \text{EM}_{cf}}{\text{EM}_{\text{clean}}} \times 100\% \quad \downarrow. \tag{3}$$

A lower ADR indicates that the model retains a greater portion of its clean-data competence when retrieval is corrupted.

**Counterfactual Sensitivity Score (CSS).** Beyond surface-level accuracy, we also probe whether the model's decision process is destabilized by adversarial edits. CSS measures the aggregate change in answer-token log-probabilities between clean and counterfactual contexts:

$$\Delta\text{CSS} = \sum_{t \in \text{Ans}} \left| \log p_{\text{clean}}(t) - \log p_{\text{cf}}(t) \right| \quad \downarrow. \tag{4}$$

Large CSS values mean the model's confidence swings sharply once retrieval is corrupted, whereas smaller values correspond to steadier reasoning.

**Parametric Knowledge Score (PKS).** Following Sun et al. (2025), we probe how much each transformer block relies on its parametric memory. Each block integrates two flows: (i) the residual stream, carrying context from the prompt (including retrieval), and (ii) the feed-forward network (FFN), injecting stored knowledge from the model's parameters. PKS quantifies how strongly the FFN reshapes token-level logits. For layer $\ell$ and answer token $t$:

$$\text{PKS}_{\ell,t} = \text{JSD}\big(\text{softmax}\big(W_U \ \text{LN}(h_{\ell,t}^{\text{mid}})\big), \text{softmax}\big(W_U \ \text{LN}(h_{\ell,t}^{\text{out}})\big)\big), \tag{5}$$

where $h^{\text{mid}}$ is the hidden state before the FFN, and $h^{\text{out}}$ is after adding the FFN output back to the residual path.

1. A **small PKS** means the FFN leaves logits nearly unchanged, passing through contextual evidence.
2. A **large PKS** means the FFN significantly overwrites logits, signaling reliance on parametric memory.

For robustness analysis, we compute the per-layer shift between clean and noisy contexts:

$$\Delta\text{PKS}_\ell \ = \ \frac{1}{|\text{Ans}|} \sum_{t \in \text{Ans}} \big(\text{PKS}_{\ell,t}^{\text{cf}} - \text{PKS}_{\ell,t}^{\text{clean}}\big), \tag{6}$$

A positive $\Delta\text{PKS}_\ell$ indicates that, under retrieval corruption, layer $\ell$ compensates by injecting more stored knowledge, while values close to zero signal steady reliance. Plotting the curve $\{\Delta\text{PKS}_\ell\}_{\ell=1}^{L}$ provides a fine-grained, layer-wise view of how noise shifts the balance between external evidence and internal memory.

All significance tests are conducted with a paired two-tailed t-test at $p < 0.05$.

## A.4 Code and Data Release

For reproducibility, we provide an archive that contains all code, prompt templates, and processed datasets used in this work. The compressed package is available at the following anonymous Google Drive link: `https://drive.google.com/drive/folders/11sjxmYLN_vlXxmIGMGnvjtdn39tJhvir?usp=sharing`

This archive allows reviewers to fully reproduce our experiments under the same settings described in the paper.

# B  ADDITIONAL ANALYSES

## B.1  OUTCOME-CONDITIONED PKS ANALYSIS

To complement the aggregate view in Figure 5, we analyze how $\Delta$PKS differs between correct and incorrect predictions.

**IDEAL-RAG.** As shown in Figure 6, incorrect answers are strongly associated with large $\Delta$PKS spikes, while correct predictions cluster around much smaller shifts. This suggests that abrupt amplification of the knowledge-FFN pathway is a reliable indicator of error, whereas stable PKS corresponds to dependable reasoning.

**InstructRAG.** In contrast, Figure 7 reveals little separation between correct and incorrect predictions. Both groups exhibit overlapping and volatile $\Delta$PKS profiles, especially in deeper layers, with sharp spikes that appear inconsistently. This pattern reflects an uncalibrated fallback to memory: without explicit extraction or reconciliation, the model intermittently amplifies parametric pathways, sometimes helping, sometimes hurting.

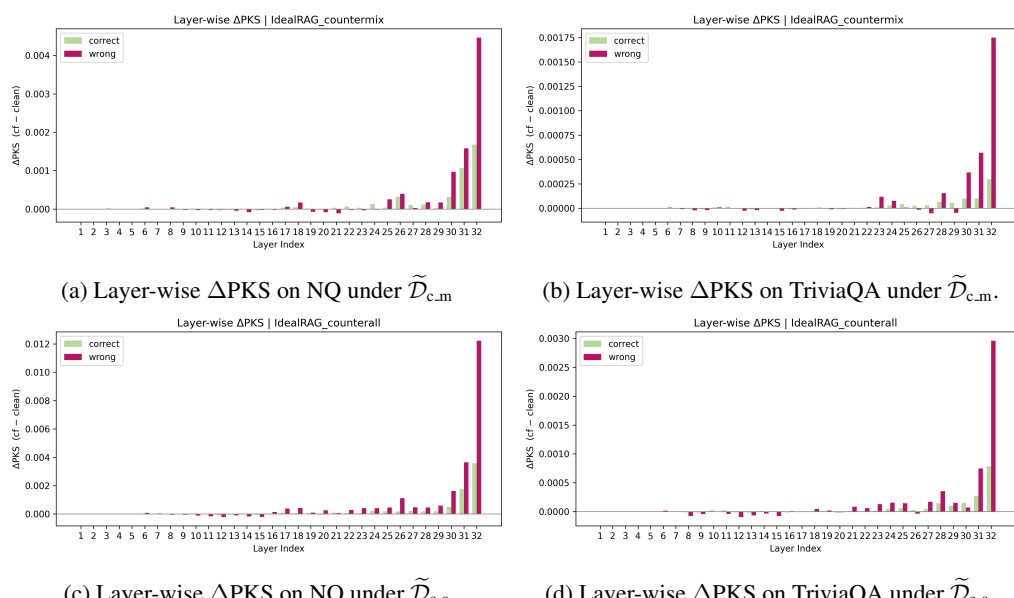

(a) Layer-wise $\Delta$PKS on NQ under $\widetilde{\mathcal{D}}_{c\_m}$

(b) Layer-wise $\Delta$PKS on TriviaQA under $\widetilde{\mathcal{D}}_{c\_m}$.

(c) Layer-wise $\Delta$PKS on NQ under $\widetilde{\mathcal{D}}_{c\_a}$

(d) Layer-wise $\Delta$PKS on TriviaQA under $\widetilde{\mathcal{D}}_{c\_a}$.

Figure 6: (green = answers correct, red = answers wrong) further shows that within IDEAL-RAG itself, questions it fails on are accompanied by a sharper late-layer $\Delta$PKS surge—especially in the final two blocks—whereas successful cases keep the rise modest. Hence a sudden, large jump in parametric logits is a reliable warning signal for impending hallucination under both the $50\%$-mix and full-corruption settings on Natural Questions and TriviaQA. (Layout and subplot lettering follow Fig 5 for visual consistency.)

**Interpretation.** These results reinforce the aggregate findings. IDEAL-RAG, by proactively surfacing parametric knowledge, stabilizes the residual stream and makes $\Delta$PKS a meaningful error signal. InstructRAG, by contrast, reacts unpredictably, aligning with Sun et al. (2025)'s observation that uncontrolled late FFN dominance is strongly tied to hallucinations.

## B.2  ANSWER-CONTAINMENT ANALYSIS

To examine when parametric knowledge is most useful, we partition the clean test questions into three categories: those where the gold answer appears only in the model's parametric memory (**inter_only**, $\mathcal{D}_{in}$), those where it is found only in the retrieved passages (**exter_only**, $\mathcal{D}_{ex}$), and those where both sources contain the answer (**both_contained**, $\mathcal{D}_{both}$). As shown in Table 6, IDEAL-RAG consistently outperforms InstructRAG in the inter_only and both_contained settings, while showing a slight deficit in the exter_only case where reliance on retrieval is unavoidable. This demonstrates

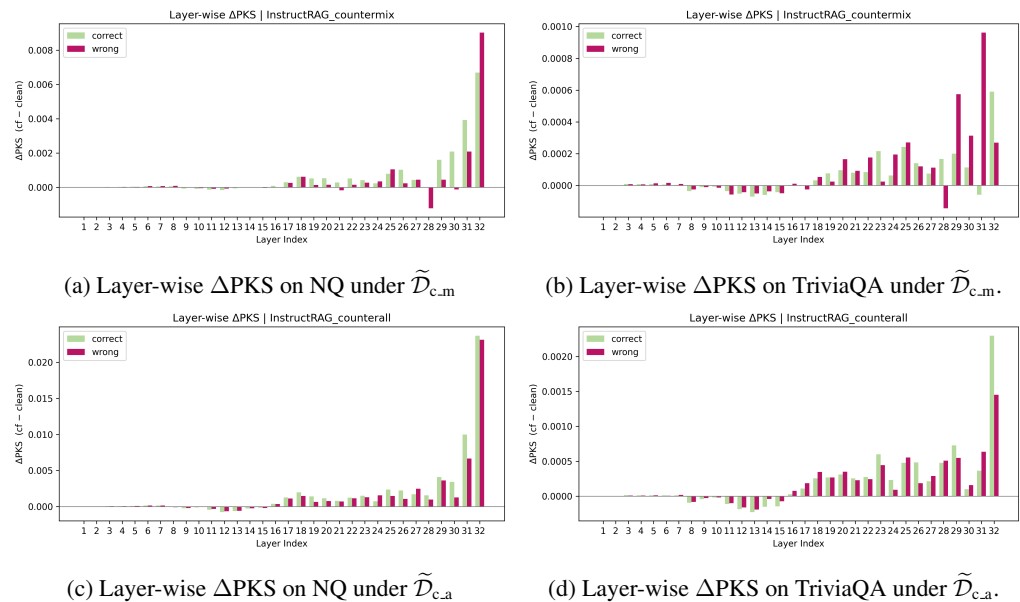

(a) Layer-wise $\Delta$PKS on NQ under $\widetilde{\mathcal{D}}_{\text{c\_m}}$

(b) Layer-wise $\Delta$PKS on TriviaQA under $\widetilde{\mathcal{D}}_{\text{c\_m}}$.

(c) Layer-wise $\Delta$PKS on NQ under $\widetilde{\mathcal{D}}_{\text{c\_a}}$

(d) Layer-wise $\Delta$PKS on TriviaQA under $\widetilde{\mathcal{D}}_{\text{c\_a}}$.

Figure 7: (green = answers correct, red = answers wrong) shows little separation between outcomes: both correct and wrong cases exhibit erratic, late-layer spikes with wide overlap and no stable trend, under both the 50%-mix and full-corruption settings on Natural Questions and TriviaQA. This pattern suggests $\Delta$PKS is a weak diagnostic for this one-shot regime—reflecting reactive, unstable use of parametric memory rather than a structured fallback. (Layout and subplot lettering match Fig. 5.)

that IDEAL-RAG excels precisely in the scenarios it was designed for—leveraging internal knowledge when external evidence is incomplete or misleading.

Table 6: For each datasets we report exact-match accuracy when the gold answer appears only in the model's parametric memory ($\mathcal{D}_{\text{in}}$), only in the retrieved passages ($\mathcal{D}_{\text{ex}}$), or in both sources ($\mathcal{D}_{\text{both}}$).

| Method | PopQA | | | NQ | | | TriviaQA | | | MultiHopQA | | |
|---|---|---|---|---|---|---|---|---|---|---|---|---|
| | $\mathcal{D}_{\text{in}}$ | $\mathcal{D}_{\text{ex}}$ | $\mathcal{D}_{\text{both}}$ | $\mathcal{D}_{\text{in}}$ | $\mathcal{D}_{\text{ex}}$ | $\mathcal{D}_{\text{both}}$ | $\mathcal{D}_{\text{in}}$ | $\mathcal{D}_{\text{ex}}$ | $\mathcal{D}_{\text{both}}$ | $\mathcal{D}_{\text{in}}$ | $\mathcal{D}_{\text{ex}}$ | $\mathcal{D}_{\text{both}}$ |
| *w/o Training* | | | | | | | | | | | | |
| InstructRAG | 36.36 | **86.76** | 94.54 | 35.42 | **76.25** | 91.80 | 59.28 | **89.27** | 96.69 | 78.44 | **53.57** | 90.21 |
| **IDEAL-RAG** | **54.55** | 82.34 | **95.45** | **71.88** | 70.07 | **93.03** | **81.01** | 78.66 | **97.21** | **86.11** | 36.88 | **91.06** |
| *w/ Training* | | | | | | | | | | | | |
| InstructRAG* | 40.91 | **89.64** | 94.77 | 39.58 | **82.83** | 92.75 | 64.03 | **90.40** | 97.37 | 81.36 | **68.36** | 93.92 |
| **IDEAL-RAG** | **63.64** | 86.69 | **96.82** | **71.88** | 70.07 | **93.03** | **83.23** | 78.19 | **97.83** | **90.29** | 44.41 | **94.01** |

## B.3 ATTENTION-DISTRIBUTION PROBE

To probe the model's decision focus, we aggregate token-level self-attention across all decoder layers and heads, then sum the weights per retrieved passage $p_j$:

$$\text{AttnScore}(p_j) = \sum_{\ell,h} \sum_{t \in p_j} \text{softmax}\big(A_{\ell,h}\big)_t, \tag{7}$$

where $A_{\ell,h}$ is the raw attention matrix at layer $\ell$, head $h$.

Figure 8 compares attention patterns under $\widetilde{\mathcal{D}}_{\text{c\_m}}$. InstructRAG often fixates on a few passages, ignoring some that contain the gold answer—explaining its fragility under counterfactual edits. IDEAL-RAG distributes attention more evenly and assigns weight to both true-answer and corrupted passages, enabling explicit comparison. This aligns with the design of the Linked Rationale module and further corroborates IDEAL-RAG's resilience to retrieval noise.

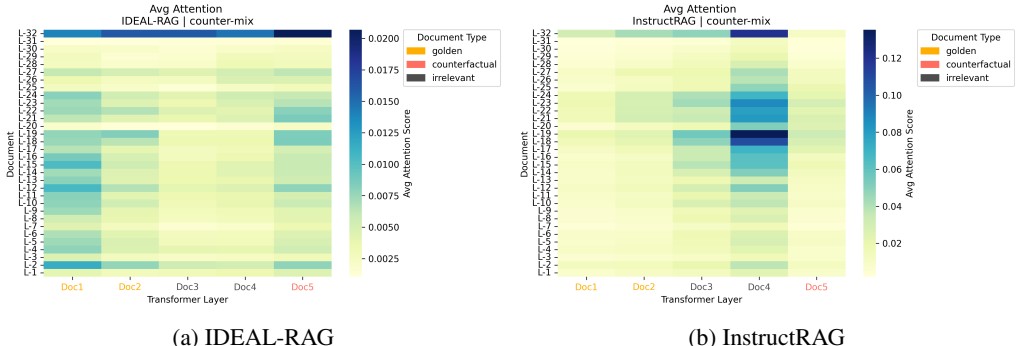

(a) IDEAL-RAG                                  (b) InstructRAG

Figure 8: Passage-level attention heat-maps on a counter-mix example. IDEAL-RAG spreads attention across all passages and highlights both the gold-answer and counterfactual segments, whereas InstructRAG concentrates on a single irrelevant passage.

## C  PROMPT SETTING

**Prompt mapping to pipeline steps.** For clarity, we summarize how each component of the IDEAL-RAG pipeline corresponds to specific prompt templates. During Parametric Knowledge Extraction ($\mathcal{E}_{\text{int}}$), the model elicits internal knowledge via the template in Figure 9. In Dual-Source Standpoint Generation ($\mathcal{G}$), answer-seen standpoints are constructed using Figure 10 and 11, while answer-unseen standpoints rely on Figure 12 and 13. Finally, in Linked Rationale Generation ($\mathcal{L}$), answer-seen linking is guided by Figure 14, few-shot inference at test time uses Figure 15.

---

**Input**:

Generate a document that provides accurate and relevant background knowledge related to the given question. The document should be informative and structured as if it were an excerpt from a knowledge source, without explicitly answering the question. Avoid unnecessary commentary, explanations, or direct responses. If relevant information is unavailable, state 'I don't know' without adding further speculation or context.

Question: {question}

Document: $\{K_{\text{int}}\}$

---

Figure 9: Prompts to extract internal knowledge from a frozen model

**Prompt settings for ablation conditions.** In the ablation experiments, we used simplified prompt configurations to isolate the effect of individual modules. For the variant **w/o $\mathcal{E}_{\text{int}}$ and $\mathcal{G}$**, we replicate the InstructRAG-style one-shot baseline: a single template (Figure 16) directly asks the model to read the retrieved passages, optionally reflect on background knowledge, and then justify its answer, without performing explicit parametric-knowledge extraction or generating dual standpoints. By contrast, for the variant **w/o $\mathcal{G}$**, we retain explicit parametric elicitation but remove the standpoint-generation stage, feeding the question, retrieved passages, and extracted memory directly into the fusion template (Figure 17). Apart from these structural changes, all other settings such as instruction style, exemplar count, and decoding strategy are identical to the full IDEAL-RAG pipeline, ensuring that observed differences arise solely from the missing mechanisms.

**Input**:

Read the following documents relevant to the given question: $\{question\}$
$\{retrieved\ documents\}$
Please answer the following question using external documents only, without relying on internal or prior knowledge: $\{question\}$ and explain how the proposed answer(s):$\{answers\}$ can be supported.

You are provided with a set of external documents. Base your explanation on the information found in these documents. If the documents do not reasonably support the proposed answer(s), you may instead present a more plausible answer that is supported by the documents, and explain why it fits better.
Do not refer to internal knowledge or prior facts beyond what the documents state or imply.

Your output should:

- Identify relevant factual claims from the external documents,
- Explain how those claims lead to the proposed answer, or support a better one, in a logically sound and document-grounded way.

Note that the question may be compositional and require intermediate analysis to deduce the final answer. Make sure your response is grounded and provides clear reasoning details followed by a concise conclusion.

**Output**: $\{\mathcal{S}_{\text{ext}}^{\star}\}$

Figure 10: Prompts to generate external standpoint in answer-seen scenario.

**Input**:

Read the following documents relevant to the given question: $\{question\}$
$\{extracted\ internal\ knowledge\}$
Please answer the following question using internal knowledge only, without referring to any external documents: $\{question\}$ and explain how the proposed answer(s):$\{answers\}$ can be supported.

You are provided with a set of internal knowledge statements. Base your explanation primarily on these statements. If the internal knowledge does not reasonably support the proposed answer(s), you may instead present a more plausible answer that is supported by the internal knowledge, and explain why it fits better.

Your output should:

- Identify the factual claims from internal knowledge (provided or known),
- Explain how those claims lead to the proposed answer, or support a better one, in a logically sound and verifiable way.

Note that the question may be compositional and require intermediate analysis to deduce the final answer. Make sure your response is grounded and provides clear reasoning details, followed by a concise conclusion.

**Output**: $\{\mathcal{S}_{\text{int}}^{\star}\}$

Figure 11: Prompts to generate internal standpoint in answer-seen scenario.

**Input**:

Your primary task is to answer the given question by analyzing the provided external documents. You must evaluate their relevance, accuracy, and completeness in relation to the question. If the documents clearly support a specific answer, explain how they lead you to that answer. If the documents are incomplete, ambiguous, or conflicting, make your best judgment based only on what is found in the documents. Do not use internal or prior knowledge.
Below are some examples of how to give the rationale:
$\{\mathcal{B}_{\text{ext}}\}$

Now it is your turn to analyze the following documents and answer the given question.
$\{retrieved\ documents\}$

Based on the provided information, answer the question: $\{question\}$

**Output**: $\{\hat{\mathcal{S}}_{\text{ext}}\}$

Figure 12: Prompts to generate external standpoint in answer-unseen ICL task.

**Input**:

Your primary task is to answer the given question by analyzing the internal knowledge provided. You must examine this information to determine whether it contains enough evidence to support a clear answer. If it does, explain how the internal knowledge leads to your answer. If it does not, use your broader internal knowledge to offer the most plausible answer you can, but do not use any external sources or documents.
Below are some examples of how to give the rationale:
$\{\mathcal{B}_{\text{int}}\}$

Now it is your turn to analyze the following internal knowledge and answer the given question.
$\{extracted\ internal\ knowledge\}$

Based on your internal knowledge and the provided information, answer the question: $\{question\}$

**Output**: $\{\hat{\mathcal{S}}_{\text{int}}\}$

Figure 13: Prompts to generate internal standpoint in answer-unseen ICL task.

**Input**:

Read the following documents relevant to the given question: $\{question\}$
$\{retrieved\ documents\}$
$\{extracted\ internal\ knowledge\}$
You are given the following:
Question: $\{question\}$ Correct answer(s): $\{answers\}$

Two independent arguments attempt to justify what they believe to be the correct answer.
External Standpoint: $\{\mathcal{B}_{\text{ext}}\}$
Internal Standpoint: $\{\mathcal{B}_{\text{int}}\}$

Your task is to analyze both arguments and determine how internal and external information can contribute to reasoning toward the correct answer. Rather than choosing a side, your goal is to organize the relevant reasoning from both sources, identify which parts align with the correct answer, and explain how the conclusion can be supported.

In your explanation:

- Identify the key claims made in each argument.
- Compare them against the correct answer.
- Accept claims that logically support the correct answer.
- Reject claims that are inconsistent, unsupported, or contradict the correct answer and explain why.
- Integrate useful information from both sides to construct a coherent, step-by-step explanation that leads to the correct answer.

Base your explanation only on the arguments and the known correct answer.
Note that the question may be compositional and require intermediate analysis to deduce the final answer. Make sure your response is grounded and provides clear reasoning details followed by a concise conclusion.

**Output**: $\{\mathcal{R}_{\text{link}}^{\star}\}$

Figure 14: Prompts to generate linked rationale in answer-seen scenario.

---

**Input**:

Your primary task is to answer the given question by analyzing two competing arguments, each supported by a different type of information: one by external documents, the other by internal knowledge. Each argument includes its own reasoning and its supporting source content.
You must carefully evaluate how well each argument uses its respective source to justify its answer. If one side clearly provides stronger evidence and more valid reasoning, explain why you find it more convincing. If both arguments are incomplete, ambiguous, or equally strong, make your best judgment based only on the information provided. Do not rely on outside or prior knowledge.
Below are some examples of how to give the rationale:

$\{B^\star_{\text{link}}\}$

Now it is your turn to analyze the following materials and answer the given question.
$\{retrieved\ documents\}$
$\{extracted\ internal\ knowledge\}$

Two independent arguments attempt to justify what they believe to be the correct answer.

External Standpoint: $\{\hat{\mathcal{S}}_{\text{ext}}\}$
Internal Standpoint: $\{\hat{\mathcal{S}}_{\text{int}}\}$

Based on the provided arguments and their supporting information, answer the question: $\{question\}$

**Output**: $\{\hat{\mathcal{R}}_{\text{link}}\}$

---

Figure 15: Prompts to generate linked rationale in answer-unseen ICL task.

---

**Input**:

Your primary task is to answer the given question by first reflecting on what internal knowledge you have that might be relevant, and then critically analyzing the provided documents.
You must evaluate the relevance, accuracy, and sufficiency of both internal and external information in relation to the question.
Below are some examples of how to give the rationale:

$\{B^\star_{\text{inst}}\}$

Now it is your turn to analyze the following documents and answer the given question.
$\{retrieved\ documents\}$

Based on both your internal knowledge and the provided information, answer the question:
Question: $\{question\}$

**Output**:

---

Figure 16: Prompts that refine the InstructRAG setup to generate rationales in the answer-unseen ICL task. The exemplar bank $B^\star_{\text{inst}}$ contains rationales generated under the InstructRAG method in the answer-seen setting.

**Input**:

Your primary task is to answer the given question by analyzing both the provided external documents and internal knowledge. You must evaluate how well each source contributes to answering the question, and whether they support one or more of the proposed answer labels.
Below are some examples of how to give the rationale:
$\{B^{\star}_{\text{one-step}}\}$

Now it is your turn to analyze the following materials and answer the given question.
$\{retrieved\ documents\}$

Answer a given question using the information from both externally retrieved documents and your own memorized documents.
Question: $\{question\}$

**Output**:

Figure 17: Prompts for generating linked rationales without explicit standpoints ($\mathcal{G}$) in the answer-unseen ICL setting.

## D    USE OF LARGE LANGUAGE MODELS (LLMS)

In line with the ICLR 2026 submission policy, we disclose the use of large language models (LLMs) during manuscript preparation. Specifically, we used ChatGPT (OpenAI) as a general-purpose writing assistant to refine grammar, improve readability, and polish phrasing. ChatGPT was not involved in research ideation, experiment design, data analysis, or the generation of scientific claims. All scientific content, results, and conclusions are solely the responsibility of the authors.

