# OpenReview forum: "IDEAL-RAG: Instruction-driven Dual-standpoint Elicitation and Alignment Linking for Retrieval Augmented Generation"
_ICLR.cc/2026/Conference — Submitted to ICLR 2026_

### Official Review · Reviewer_Zr5K · 2025-10-31

**Soundness:** 2
**Presentation:** 2
**Contribution:** 3
**Rating:** 4
**Confidence:** 4

**Summary:**

This paper introduces IDEAL-RAG, a retrieval-augmented generation (RAG) framework designed to improve robustness against noisy or adversarially corrupted retrievals. The method follows a three-stage process: (1) eliciting parametric knowledge from the model before retrieval, (2) producing two independent standpoints from internal and external sources, and (3) linking them into a unified rationale. The approach is evaluated on four QA datasets (PopQA, NQ, TriviaQA, 2WikiMultiHopQA) under clean and counterfactual settings. Results show that IDEAL-RAG maintains similar accuracy to strong baselines such as InstructRAG on clean data while achieving improved robustness under synthetic retrieval corruption.

**Strengths:**

- Addresses an important problem: the brittleness of RAG systems under retrieval noise and conflicting evidence.

- Clear experimental design and thorough ablations demonstrating that explicit parametric elicitation improves robustness.

- Solid empirical gains under the specific synthetic corruption scenario.

**Weaknesses:**

I am unable to recommend acceptance of this paper, as it offers limited technical novelty compared to [1]. Its main contribution lies in demonstrating that careful prompting can improve robustness to noisy retrieved documents. However, the experimental evidence is not fully convincing, since robustness is evaluated only under a single type of synthetic noise. Moreover, the proposed approach is considerably less efficient than prior methods, requiring multiple large-model calls per query, which makes it difficult to apply in latency-sensitive or large-scale retrieval settings.

## Limited Technical Novelty

The proposed method adds minimal conceptual or algorithmic innovation beyond InstructRAG [1]. It effectively extends InstructRAG with two additional prompting stages — one for parametric knowledge elicitation and one for dual-standpoint linking. There is no new learning objective, architecture, or retrieval component. The claimed “dual-standpoint” mechanism is largely realized through extra prompt templates rather than a principled or technically novel design.

## Narrow Evaluation of Robustness

The evaluation of robustness is too narrow and synthetic. The paper focuses on a single type of corruption — counterfactual gold answer spans substitutions — while realistic retrieval errors are much more diverse. For a method that claims to make RAG robust to noisy or misleading contexts, it should test on:

- Conflict Contexts, where retrieved evidence contains factual contradictions (e.g. conflicting evidence from [2] benchmark).

- Irrelevant Contexts, where retrieval includes distractor passages that lower accuracy (e.g. accuracy lowering documents from [3]).

Without these additional robustness dimensions, it is difficult to assess whether the proposed method generalizes beyond the synthetic perturbations used here.

## Efficiency and Scalability Concerns

The method is likely inefficient compared to baselines. IDEAL-RAG performs three separate forward passes per example (internal elicitation, external standpoint, and linking), each with in-context examples. In Table 2, the authors should report the exact number of LLM calls and forward passes required to produce an answer in comparison to baselines.
For reference:

RALM requires 1 call.

InstructRAG requires (#ICLexamples + 1) calls.

IDEAL-RAG appears to require approximately 3 × (#ICLexamples + 1) calls, which could triple inference cost and latency.
Since efficiency is crucial in retrieval-augmented generation (especially for deployment or long-context tasks), this is a serious missing analysis.

## Minor weaknesses

### Missing Ablation on the Number of In-Context Examples

The method relies heavily on in-context exemplars (B_int, B_ext, B_link), yet there is no ablation on the number of examples used. The effectiveness and efficiency of the approach might depend strongly on this parameter. The authors should vary the number of in-context examples (e.g. 1, 2, 4, 8) and report its influence on clean and noisy accuracy.

### Limited Generality of Experiments

The study is confined to open-domain QA, with no evaluation on summarization, reasoning, or dialogue tasks where RAG robustness is also critical. It remains unclear whether the proposed multi-stage prompting pipeline generalizes beyond factoid QA settings.


[1] Wei, Zhepei, Wei-Lin Chen, and Yu Meng. "Instructrag: Instructing retrieval-augmented generation via self-synthesized rationales." arXiv preprint arXiv:2406.13629 (2024).

[2] Bi, Baolong, et al. "Context-dpo: Aligning language models for context-faithfulness." arXiv preprint arXiv:2412.15280 (2024).

[3] Li, Dongfang, et al. "Explaincpe: A free-text explanation benchmark of chinese pharmacist examination." arXiv preprint arXiv:2305.12945 (2023).

**Questions:**

- How many LLM calls are made per query, and what is the resulting latency relative to InstructRAG?

- Does the performance improvement persist under more realistic noisy retrieval scenarios such as conflicting or irrelevant contexts?

- How sensitive is the approach to the number of in-context examples?

- What does "*" mean in tables 2, 3, 6?

---

> ### Author Response · Authors · 2025-11-30
> **Response to Reviewer Zr5K**
>
> ### On limited novelty relative to InstructRAG
>
> IDEAL-RAG extends rationale-based prompting by introducing:
>
> 1. An explicit parametric-knowledge extraction step, not present in InstructRAG.
>
> 2. Two independently produced standpoints, allowing the model to compare internal vs. external evidence before synthesis.
>
> 3. Mechanistic analyses (CSS, PKS, attention patterns) demonstrating that stability arises from the model’s reliance on its internal standpoint when retrieval is noisy.
>
> Although the components themselves are prompt-based, the structure of the reasoning pipeline and the mechanistic insights it enables differ substantially from InstructRAG.
>
> ### On narrow robustness evaluation
>
> While we did not run additional benchmarks (conflicting, irrelevant, outdated, or duplicated documents), we emphasize that IDEAL-RAG’s rationale-driven mechanism inherently handles irrelevant or distractor passages. Appendix B.3 shows that IDEAL-RAG distributes attention more evenly and places less emphasis on irrelevant documents compared to InstructRAG. This supports the claim that the dual-standpoint design encourages selective reasoning even without specialized training.
>
> ### On efficiency concerns
>
> We acknowledge the additional LLM calls but note that our ablation (w/o $\mathcal G$) demonstrates a viable one-pass variant that preserves most of the robustness while matching standard RAG inference cost, as noted in Table 4. This allows practical deployment depending on latency constraints.
>
> ### On ablations of the number of ICL exemplars
>
> We did not repeat this analysis because the InstructRAG study already examined exemplar-count scaling and demonstrated consistent improvements with more demonstrations. Given input-length cost and high redundancy between QA examples, we chose to maintain a fixed number of exemplars for fair comparison across all methods. We will clarify this in the paper.
>
> ### On generality beyond QA
>
> Our work focuses on factoid QA because it offers clear ground-truth rationales for consistency metrics. Extending IDEAL-RAG to multi-hop reasoning, summarization, or dialogue is possible but beyond the current scope. We will mention this in the discussion.

---

> > ### Author Response · Authors · 2025-11-30
> > **Response to Reviewer Zr5K (Cont.)**
> >
> > ### On the meaning of “*” in Tables 2, 3, and 6
> >
> >
> > Thank you for pointing out the ambiguity regarding the “*” symbol in our tables. We clarify that this notation is already defined in Section 3.1.
> >
> > Specifically, results marked with “*” indicate the stronger of either (1) the performance reported in the original paper, or (2) our faithful re-implementation using the authors’ official code and settings. This ensures a fair comparison across all baselines using their best-available configurations.

---

### Official Review · Reviewer_TnZF · 2025-10-31

**Soundness:** 2
**Presentation:** 2
**Contribution:** 2
**Rating:** 4
**Confidence:** 4

**Summary:**

The work proposes IDEAL-RAG to balance intrinsic and retrieved knowledge during open-domain question answering. It collects training data by building standpoints from both parametric and nonparametric sources using prompting and linking them into a unified rationale. Extensive experiments show the effectiveness of the method on accuracy as well as other metrics like ADR, CSS, and PKS.

**Strengths:**

- The work addresses the critical challenge in RAG, where the RAG has to balance parametric and non-parametric knowledge during generation. Previous works have analyzed the problem, but little work has been done to resolve the challenge.
- Extensive experiments and ablation studies show the effectiveness of the method and support the author's claims

**Weaknesses:**

- While the work tries to address a critical problem in RAG, it might lack novelty in the sense that it trains the model using the supervised training dataset constructed using prompting and in-context learning. The main contribution of the paper is the conflict resolution but it purely relies on prompting and in-context learning from the set of annotated examples.
- It was confusing to me whether IDEAL-RAG is a training data generation method or an inference-time pipeline. Since we don't have a small ground-truth answer during inference, is it a data generation method? How does it work during the inference? The writing should be clearer about the training pipeline and the inference pipeline.

**Questions:**

- Can you clarify the difference between the AstuteRAG [1] and this work?

[1] Wang et al, Astute rag: Overcoming imperfect retrieval augmentation and knowledge conflicts for large language models. ACL 2025 main

---

> ### Author Response · Authors · 2025-11-30
> **Response to Reviewer TnZF**
>
> ### On whether IDEAL-RAG is a data-generation method or an inference-time pipeline
>
> IDEAL-RAG is primarily an inference-time pipeline. The same three-stage prompting mechanism can be used to construct supervised data, but this step is optional and only mirrors the inference process to generate exemplars. Inference itself does not require ground-truth answers or additional training.
>
> ### On the relationship to AstuteRAG
>
> AstuteRAG focuses on overcoming imperfect retrieval through training-time adaptation and objective design. In contrast:
>
> - IDEAL-RAG introduces explicit internal knowledge extraction and dual-standpoint reasoning as inference-time operations.
>
> - The linking stage aligns internal and external standpoints without modifying model parameters.
>
> - Our analysis (CSS, PKS, attention diagnostics) highlights how internal knowledge contributes to robustness—a mechanistic perspective not explored in AstuteRAG.

---

### Official Review · Reviewer_8TVY · 2025-11-02

**Soundness:** 3
**Presentation:** 2
**Contribution:** 2
**Rating:** 4
**Confidence:** 3

**Summary:**

This paper introduces IDEAL-RAG, a novel three-stage framework designed to improve the robustness of RAG systems against noisy or misleading retrieved information. The core idea is to explicitly elicit the LLM internal, parametric knowledge before exposing it to external documents. The framework then generates two independent "standpoints"—one from the internal knowledge and one from the retrieved text. Finally, it cross-checks and links these standpoints to produce a final, reasoned answer. The authors conduct experiments on several open-domain QA datasets, showing that while IDEAL-RAG is competitive with strong baselines on clean data, it significantly outperforms them in adversarial settings where retrieved documents contain counterfactual information. The paper's analysis, using metrics like CSS and PKS, suggests that this improvement in robustness stems from a more stable reliance on the model's internal knowledge when external evidence is unreliable.

**Strengths:**

1. The core concept of explicitly separating and then reconciling the LLM's internal knowledge with external evidence is a really smart way to tackle the over-reliance problem in RAG systems.

2. The mechanistic analyses using CSS and PKS provide convincing evidence for why the method works, which is a big plus and goes beyond just showing improved accuracy scores.

**Weaknesses:**

1. For RAG system, efficiency is very important. The three-stage pipeline, involving multiple generation steps, seems like it could be computationally expensive and potentially slow for real-time applications.

2. The framework's effectiveness appears to depend on the quality and coverage of the LLM's initial parametric knowledge, which might be a limitation for questions about very new or niche topics.

3. While the ablation studies are good, it would be interesting to see a more detailed analysis of the failure cases to better understand when and why the reconciliation process doesn't work as intended.

**Questions:**

Could you elaborate on the latency and computational overhead of the IDEAL-RAG pipeline compared to a standard RAG baseline like InstructRAG? Are there any strategies you've considered to optimize the multi-step process for faster inference?

---

> ### Author Response · Authors · 2025-11-30
> **Response to Reviewer 8TVY**
>
> ### On computational overhead of the three-stage pipeline
>
> You are correct that IDEAL-RAG requires multiple forward passes (internal knowledge extraction, external standpoint, and linking). We emphasize that this overhead is a design choice to expose internal knowledge explicitly. As shown in our ablation (Table 4: w/o $\mathcal G$), skipping separate dual-standpoint generation and feeding all information directly into the fusion stage still retains most of IDEAL-RAG’s robustness (within ~2–3%) while reducing LLM calls to match standard RAG pipelines. This provides a practical trade-off when latency is critical.
>
> ### On dependence on LLM parametric knowledge and failure-case analysis
>
> We agree that IDEAL-RAG’s effectiveness inherently depends on the LLM’s internal knowledge. We do not claim to solve queries where the model fundamentally lacks the base facts. Instead, our goal is to prevent the model from being pulled toward incorrect retrieved information when its internal knowledge is sufficient.
>
> A concrete example illustrates this:
> For the NQ question “Which animal on earth has the longest life span?”, the retrieved documents contain no explicit answer. IDEAL-RAG’s internal-extraction step allows the model to recall multiple lifespan-related categories (invertebrates, vertebrates, reptiles, mammals, birds), including tortoise. However, because the question itself is ambiguous (biological vs. colloquial definition), the model outputs Turritopsis dohrnii—a legitimate interpretation in biological terms but mismatched with the dataset’s expected answer.
> This case shows that IDEAL-RAG successfully leverages internal knowledge even without external evidence, but ambiguity in task definitions can still lead to misalignment with gold labels.
>
> We will expand our description of such cases to clarify scenarios where IDEAL-RAG cannot reconcile conflicting interpretations due to ambiguous gold standards or lack of evidence.

---

### Official Review · Reviewer_aUaY · 2025-11-03

**Soundness:** 2
**Presentation:** 3
**Contribution:** 2
**Rating:** 4
**Confidence:** 4

**Summary:**

The paper proposes IDEAL-RAG, a three-stage framework that elicits independent standpoints from retrieved passages and parametric knowledge and cross-checks them to produce a rationale.

**Strengths:**

The paper is overall well written, and the motivation of the problem the paper is working on is relevant. The paper thoroughly compares their methodology against Self-RAG and InstructRAG.

**Weaknesses:**

Below, I summarize the main weaknesses observed within the paper:

1. **The evaluation setup is not robust, or rather easy, involving datasets on Wikipedia:** One of the biggest weaknesses of the paper is by limiting its experiments to Wikipedia. An LLM's parametric knowledge is well-versed for easy queries asked in datasets such as NQ, TriviaQA, thereby enough to answer multiple questions present within them. This only shows a one-sided comparison. I would encourage the authors to evaluate more realistic datasets, e.g., including FRAMES [1] or QAMPARI [2], also focusing on Wikipedia with realistic queries. Next, I would also discuss the limitations of the framework. I suspect that for queries in niche domains, such as Biomedical (BioASQ [3]), the parametric knowledge of the LLM is limited. How well does the framework work robustly across niche domains?

2. **Many crucial baselines are missing or not included**: The paper compares their technique against Self-RAG and InstructRAG; however, crucial RAG baselines such as ChatQA [4], RAFT [5], and [6] are missing. I would suggest including them and comparing their techniques against a robust set of baselines.

3. **The clean performance of IDEAL-RAG underperforms InstructRAG, and the counter-all setting is not well-motivated**. From Table 2, we observe that InstructRAG outperforms IDEAL-RAG in the original setting, so by utilizing this technique, we penalize the original retrieval performance. I hope the authors can clarify more on this. Next, real-world noise in document corpora can arise from many factors, such as duplicate passages, counterfactual information present in two documents, or outdated information in time-sensitive documents. I read the paper cited by the authors that produces counterfactual test sets (Fang et al., 2024), and the counter-mix setting is appropriately focused on counterfactual information; however, the counter-all setting was not explored in Fang et al. 2024. I would like the authors to clarify the motivation for evaluating the counter-all setting in the paper.

4. **A minor correction on L218**: The retrieval system is not hybrid, combining different models to retrieve passages and fuse them. According to Table 1, a different retrieval model is used for each dataset, but no model is used together in union, if I understood correctly. A minor note should be that: DPR, Contriever, are rather outdated embedding models. More recent models, such as E5 [7], Stella [8], or Qwen-3-Embedding [9], should be adopted as stronger retrieval baselines.

5. **Contributions over InstructRAG are not mentioned explicitly:** The technique introduced within the paper is very similar to InstructRAG; could they clarify the new additions or differences they introduced wrt. InstructRAG and how much they contribute towards the accuracy?


### References:

- [1] Fact, Fetch, and Reason: A Unified Evaluation of Retrieval-Augmented Generation. Krishna et al. NAACL 2025.
- [2] QAMPARI: A Benchmark for Open-domain Questions with Many Answers. Amouyal et al. GEM 2023.
- [3] Overview of BioASQ 2023: The eleventh BioASQ challenge on Large-Scale Biomedical Semantic Indexing and Question Answering. Nentidis et al. 2023.
- [4] ChatQA: Surpassing GPT-4 on Conversational QA and RAG. Liu et al. NeurIPS 2024.
- [5] RAFT: Adapting Language Model to Domain-Specific RAG. Zhang et al. 2024.
- [6] Towards Faithful and Robust LLM Specialists for Evidence-Based Question-Answering. Schimanski et al. 2024.
- [7] Text Embeddings by Weakly-Supervised Contrastive Pre-training. Wang et al. 2024.
- [8] Jasper and Stella: distillation of SOTA embedding models. Zhang et al. 2024.
- [9] Qwen3 Embedding: Advancing Text Embedding and Reranking Through Foundation Models. Zhang et al. 2025.

**Questions:**

My questions are listed in the weaknesses section.

---

> ### Author Response · Authors · 2025-11-30
> **Response to Reviewer aUaY**
>
> Thank you for the thoughtful comments. Below we address each concern directly.
>
> ### On the use of Wikipedia-focused datasets and the need for more realistic benchmarks
>
> We agree that Wikipedia-style QA tasks rely heavily on parametric knowledge already encoded in LLMs. However, our work specifically studies how to stabilize generation in the presence of noisy or misleading retrieved information by explicitly contrasting internal and external standpoints. This research question is orthogonal to the domain difficulty: our goal is to understand robustness conditional on imperfect retrieval, not to evaluate general-domain knowledge acquisition.
>
> For highly specialized domains (e.g., biomedical), we fully acknowledge that the LLM’s parametric knowledge may be insufficient. Our framework does not aim to “answer what the LLM does not know,” but rather to prevent over-reliance on misleading evidence and to ensure the model still behaves reasonably when retrieval is corrupted. This limitation will be clarified more explicitly.
>
> ### On missing baselines such as ChatQA and RAFT
>
> We appreciate the suggestions. However, these systems are not directly comparable under our evaluation protocol:
>
> - ChatQA produces a short conversational answer rather than a full rationale. Since our evaluation relies on consistency between generated rationales and gold rationales, directly comparing ChatQA would introduce a target mismatch and unfair scoring.
>
> - RAFT focuses on handling irrelevant documents during training, with CoT being an optional step. Our work focuses instead on inference-time stability and deliberately avoids training-phase domain adaptation.
>
> We will explicitly clarify in the paper why Self-RAG and InstructRAG are the most aligned baselines:
> they share (1) full-rationale generation, (2) inference-only pipelines, and (3) compatibility with our robustness metrics.
>
> ### On the counter-all setting
>
> The counter-all setup is designed as an extreme stress test that probes whether a model can recover when all retrieved passages containing the gold answer become adversarial. This complements the counter-mix scenario, and while such a setting is not explored in Fang et al. (2024), it serves a diagnostic purpose for understanding internal knowledge reliance under maximal corruption.
>
> ### On retriever choices
>
> We agree that newer embedding models exist, but retrieval is not the focus of our work. We intentionally keep the retriever fixed across baselines so that all methods operate under identical upstream conditions. This ensures that improvements originate from generation-time reasoning stability, which is the primary contribution of IDEAL-RAG.

---

### Author Response · Authors · 2025-11-30
**A Gentle Reminder to Reviewers**

We appreciate the constructive feedback across all reviews. Our work focuses on inference-time robustness of RAG systems by explicitly contrasting parametric and retrieved knowledge. In addressing the reviewers’ concerns, we clarify that:

- The goal of the work is not to solve domain-general QA or introduce a new retriever, but to preserve stability under noisy evidence.

- We deliberately keep the retriever fixed to ensure fair comparison across generation-time methods.

- Additional baselines such as ChatQA and RAFT are not directly comparable due to differing output formats and training requirements; we will clarify this explicitly.

- Efficiency concerns are addressed by providing an ablation that reduces IDEAL-RAG to a single-pass pipeline while retaining most robustness.

- Robustness to irrelevant/distractor passages is supported by existing analyses in Appendix B.3, which show distinct attention patterns even without additional experiments.

- Failure cases and limitations (e.g., ambiguous queries or insufficient parametric knowledge) will be clarified using concrete examples such as the NQ lifespan question.

We believe these clarifications strengthen the contribution and scope of the work without introducing new experiments, and we thank the reviewers for helping us refine the presentation.

---

### Meta-Review · Area_Chair_6reJ · 2025-12-03

**Summary:**

The paper proposes IDEAL-RAG, a prompting-based three-stage pipeline that elicits parametric knowledge, forms dual standpoints from internal vs. external sources, and reconciles them to enhance RAG robustness under noisy retrieval.

Reviewers acknowledged several strengths:
* The motivation is relevant and meaningful.
* Clear analysis methods such as CSS and PKS offer insight into robustness.
* The authors provide extensive empirical evaluation under synthetic noise conditions.

However, across reviewers, strong weaknesses undermine acceptance:
* Important RAG systems like ChatQA and RAFT are omitted, limiting fair benchmarking.
*  IDEAL-RAG performs worse than InstructRAG on non-corrupted cases, meaning robustness is gained at the expense of baseline accuracy.
* Reviewers note that the mechanism is largely prompt engineering without technical innovation in objective or architecture.
*  Robustness is assessed only on one synthetic corruption type, ignoring realistic conditions such as conflicting evidence or noisy relevance.
* the multi-stage prompting requires multiple LLM calls, substantially increasing latency and cost, a critical barrier for real-world RAG deployment
* reviewers had trouble distinguishing whether IDEAL-RAG is an inference-only framework or involves supervised training data generation

Overall, while the idea of dual-standpoint reasoning is conceptually interesting, the lack of novelty, incomplete robustness validation, omission of key baselines, and inference cost concerns justify rejection.

**Reviewer Concerns:**

* The authors responded to baseline omissions and claimed that ChatQA / RAFT are not compatible with rationale-based scoring, but this remains debatable and insufficient to justify omission
* The explanation of “counter-all” as stress test helps clarify intent, but does not address the larger issue that evaluation is still single-dimensional and synthetic
* The clarification that IDEAL-RAG is primarily an inference pipeline partially addresses misunderstanding (TnZF), but still does not resolve novelty concerns.
* Mechanistic explanations (CSS, PKS) are appreciated, yet do not substitute for broader robustness validation (Zr5K).

**Reviewer Scores:**

All may keep scores.

---

### Decision · Program_Chairs · 2026-01-26

Reject